# Brief Communication: Daily, gap-free snow cover information based on a combination of NPP VIIRS and MODIS data

Andreas J. Dietz[1], Sebastian Roessler[1]

[1]German Remote Sensing Data Center (DFD), German Aerospace Center (DLR), Muenchener Strasse 20, 82234 Wessling, Germany

*Correspondence to*: Andreas Dietz (Andreas.Dietz@dlr.de)

**Abstract.** Combining Moderate Resolution Imaging Spectroradiometer (MODIS) and Visible Infrared Imaging Radiometer Suite (VIIRS) snow cover data and applying cloud- and data gap interpolation steps can be utilized to generate daily, gap-free snow cover information. Provided by the German Aerospace Center (DLR) under the name "Global SnowPack", this product has undergone several improvements, comprising the inclusion of VIIRS, a new threshold for the Normalized Difference Snow index (NDSI), and considerable validation. The Global SnowPack offers unique opportunities for analyzing time series of snow cover data since September 2000 and is freely available for visualization and download from DLRs GeoService.

## 1 Introduction

The German Remote Sensing Data Center (DFD) of DLR has released a globally available, cloud- and gap-free snow cover product in 2015 (Dietz et al., 2015) based on the MODIS Snow Cover Daily L3 Global 500 m SIN Grid product (Hall and Riggs, 2021). Named "Global SnowPack" (GSP) and being made available on the DLR Geoservice[1], several combination and interpolation steps are applied to the original snow cover product of the National Snow and Ice Data Center (NSIDC) to ensure continuous information about the presence or absence of snow cover throughout the year. The motivation for this development was the need for continuous, gap-free data for the detection of trends in snow cover duration for hydrological studies (Roessler et al., 2021; Vydra et al., 2024) or analyses in the context of animal migration, long-term changes in animal behaviour (Pokrovsky et al., 2024), or even the impact of snow cover change on genetical, evolutionary aspects (Mills et al., 2018; Zimova et al., 2022). An estimate of the snow cover status below clouds is necessary for such studies, even though they come at the price of slightly reduced accuracy when compared to clear-sky conditions (Gafurov and Bárdossy, 2009).

While the initial GSP product was derived from Aqua and Terra MODIS only, in this communication we present an updated version of this approach, incorporating National Polar-orbiting Partnership (NPP) VIIRS data, resampling the spatial resolution to 371 m, and applying an adjusted Normalized Difference Snow Index (NDSI) threshold for the classification of the binary snow/no snow information that constitutes the output of the processing chain. An extensive accuracy assessment has also been conducted. The difference between GSP and alternative products such as the MODIS gap filled MOD/MYDD10A1F or the

---

[1] https://geoservice.dlr.de/web/maps/eoc:gsp:daily

Interactive Multisensor Snow and Ice Mapping System (U.S. National Ice Center, 2004) is the higher spatial resolution (371 m), the combination of both MODIS and the VIIRS sensor data, and the different interpolation of the remaining cloud/data gaps through several steps (see Dietz et al., 2015 for more details).

## 2 Data and Methodology

The snow cover products based on MODIS (MOD10A1/MYD10A1, (Hall and Riggs, 2021)) and NPP VIIRS (Riggs and Hall, 2020) constitute the basis for the daily snow cover information. These datasets come with pre-processed cloud masks, quality layers, and an NDSI layer. Landsat 5, 7, and 8 (Collection 2 Level-2 Tier 1 Surface Reflectance) data was used to calibrate the NDSI threshold to classify binary snow/no snow information and validate the final product. A total of 381 Landsat scenes with snow cover are classified following the procedure described in (Koehler et al., 2022), which is modified based on the final deliverable of the Satellite Snow Product Intercomparison and Evaluation Exercise (SnowPEx) (Ripper et al., 2019). Additionally, the 7.5 arc-second version of the Global Multi-resolution Terrain Elevation Data 2010 (GMTED2010) digital elevation model (DEM), resampled to 12 arc-seconds using bilinear resampling, is used for the processing (Danielson and Gesch, 2011).

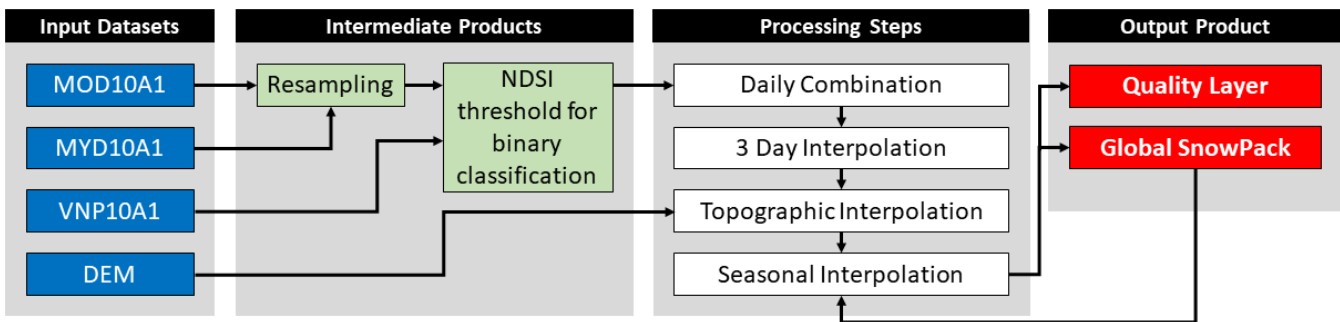

Figure 1: Workflow for the DLR Global SnowPack processing chain

A workflow has been implemented to completely remove all data gaps through a series of interpolation steps applied to the MODIS and VIIRS snow cover data. These interpolation steps (Figure 1) have been described by (Dietz et al., 2015; Gafurov and Bárdossy, 2009) and remain the same for the update presented here, with the addition of the daily VIIRS data since 2012 being an exception. Including an additional data source allows for three datasets going into the "Daily Combination" after 2012, increasing the amount of available data but not changing the general concept of the interpolation chain. There are two additional changes to the processing, regarding the spatial resolution and binary snow cover classification.

## 2.1 Adjustment of the NDSI threshold to classify snow

The data sets from MODIS (Terra: MOD10A1; Aqua: MYD10A1) and VIIRS (VNP10A1) consist of the same layers, of which "NDSI_Snow_Cover" and "NDSI_Snow_Cover_Basic_QA" are being used. The latter indicates the quality of the values determined and serves as a first orientation for the composition of the daily snow cover extent based on the combination of all

available observations; if the quality values are the same, an empirically determined order of prioritization is used. Because a binary snow cover classification is desired for purposes such as the calculation of snow cover duration or trends, a threshold for the NDSI is required to separate snow-covered ($\geq$ 50% Fractional Snow Cover; FSC) and snow-free (< 50% FSC) surfaces. The most suitable NDSI has been chosen through comparison with 381 classified Landsat reference scenes (distributed worldwide and covering all ecozones and seasons), by maximizing the F1 value. For MODIS the best NDSI threshold is an average of 0.26 (MOD10A1) and 0.23 (MYD10A1) for Aqua and Terra, respectively. For VIIRS (VNP10A1) the best threshold is 0.23.

## 2.2 Resampling of the spatial resolution to 371 m

The higher spatial resolution of VIIRS with 12 arc seconds (371 m) compared to MODIS with 15 arc seconds (463 m) made a spatial adjustment of the MODIS scenes necessary. For this purpose, the data of the "NDSI_Snow_Cover" layer was divided into a continuous and a discrete part. In the continuous part (less than or equal to 1.0) the new grid was interpolated linearly. For the discrete part (which includes classes such as water, clouds, etc.) the nearest neighbour method was used for interpolation. A comparison with the Landsat reference datasets showed that there was no significant loss of accuracy.

## 2.3 Quality layer generation

The accuracy of the data gap interpolation is mainly driven by the interpolation method and the data gap duration. Extensive tests have been conducted, simulating clouds in otherwise cloud-free sections of snow cover datasets, and removing those clouds again with the GSP algorithm. Analysing the statistics of these tests allowed conclusions about the average impact of the interpolation techniques on the accuracy. For the GSP quality layer, a value is therefore assigned based on the applied interpolation and data gap duration.

## 3 Results and Discussion

The processing chain illustrated in Figure 1, which includes the adjustments described in sections 2.1 and 2.2, leads to a binary, daily snow cover information for the whole globe since September 1[st] 2000 with a spatial resolution of 371 m. All data gaps have been interpolated applying one of the processing steps illustrated in Figure 1 and described in (Dietz et al., 2015). Although the accuracy of these steps has been assessed in earlier studies (Dietz et al., 2015; Gafurov and Bárdossy, 2009), the quality has been re-evaluated using the data from 381 Landsat classifications and 50160 globally distributed snow stations from the Global Historical Climatology Network (Menne et al., 2012). This resulted in almost 73 Mio. single measurements (with 33.4 Mio. from interpolated pixels).

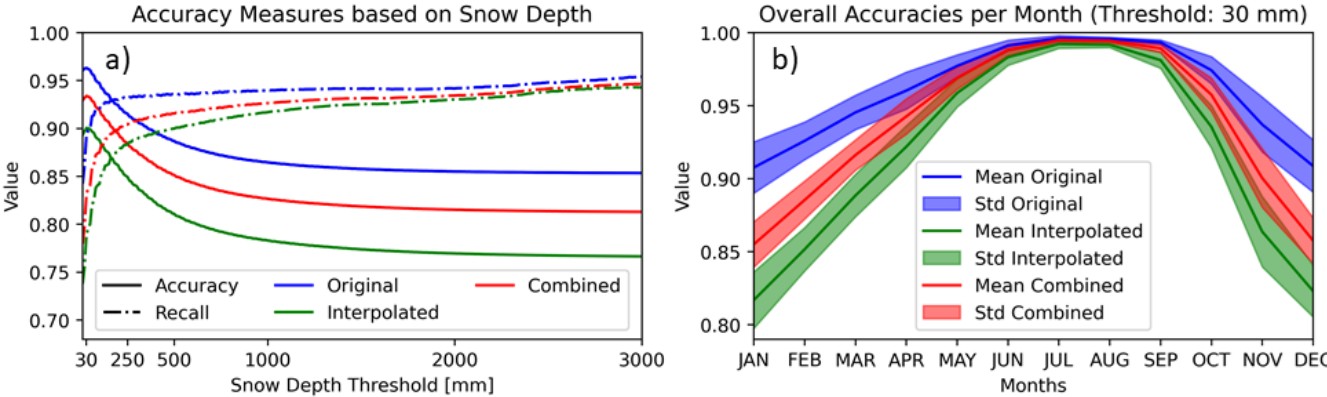

**Figure 2: Original MOD10A1/VIIRS pixels (original), the interpolated pixels (interpolated), and the combined Global SnowPack product (combined) compared to different snow depths (a); Overall accuracy of the original, interpolated, and combined pixels throughout the year, based on station data with 30 mm snow depth threshold (b).**

The challenge for the Landsat comparison is that only those pixels flagged as data gap in MODIS/VIIRS, but cloud-free in Landsat, can be compared to assess the accuracy of the interpolated pixels. The more precise cloud mask as well as the different observation time of Landsat compared to MODIS/VIIRS however allowed for a thorough assessment of almost 1 Mio. Km². The Landsat tiles were distributed all over the globe, covering North and South America, Eurasia, Australia and New Zealand, while also covering different years, seasons, land cover, and elevation zones. The comparison of the GSP with Landsat resulted

in an F1 score of 0.94. The comparison with the station data is visualized in Figure 2: The overall accuracy for the interpolated pixels reaches 90% at 30 mm snow depth, and the combined GSP product reaches values of around 93% (Figure 2a). This accuracy varies throughout the year, with values ranging from 85% to 99% for the dataset combined from the interpolated and original MOD10A1/VIIRS data (at 30 mm snow depth, Figure 2b).

The improvement of adding VIIRS to the processing chain is illustrated in Figure 3: a considerable amount of cloud-free pixels

was incorporated into the combination of daily datasets with up to an additional 10.0% (MIN 2.1%, MEAN 5.6%, MAX 10.0%) of land pixels during summer and 8.7% (MIN 0.0%, MEAN 3.3%, MAX 8.7%) of snow pixels during spring.

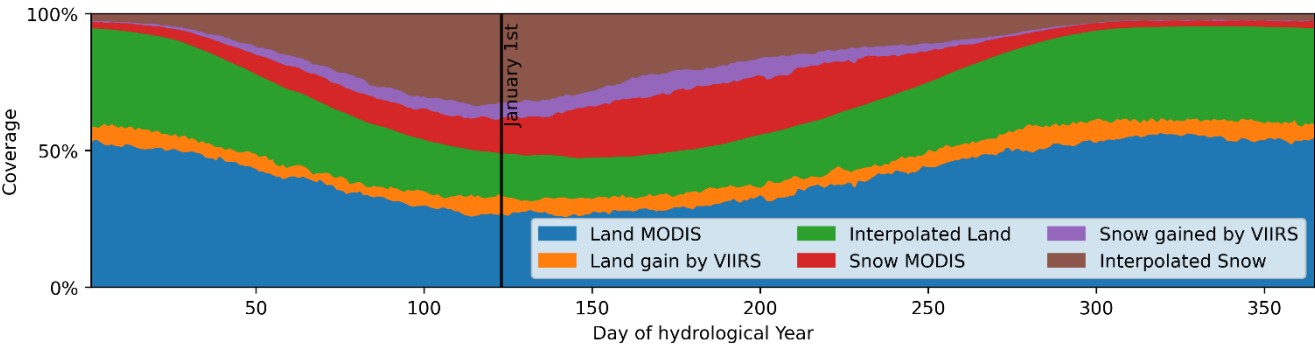

**Figure 3: Overview of the effect of the added VIIRS data: cloud-free land in MODIS (blue) and VIIRS (yellow), interpolated land (green), cloud-free snow in MODIS (red) and VIIRS (violet), interpolated snow (brown). Hydrological year runs from September 1st through August 31st.**

The initial data sources (MODIS and VIIRS snow cover products) have been assessed multiple times throughout the last decades (e.g. in (Baker and Kilcoyne, 2011; Brubaker et al., 2005; Hall et al., 2002; Hall and Riggs, 2021)), which is why the accuracy of those pixels not flagged as cloud/polar darkness or data gap were only validated with the snow depth station data (Figure 2 a) and b), referred to as "original data").

**4 Conclusions**

The updated version of the GSP is processed from three individual daily snow cover datasets (Terra and Aqua MODIS as well as NPP VIIRS) and provides a binary information about the presence or absence of snow for the whole globe. The assessment relying on Landsat data showed that resampling the MODIS 500 m data to 371 m to match VIIRS does not significantly reduce the accuracy, while the interpolation of cloud and polar darkness gaps introduces uncertainties based on the duration of the

110 data gaps. These uncertainties can easily be estimated and included in an uncertainty layer accompanying the GSP product. The accuracy assessment relying on station and Landsat data revealed an F1 score of 0.94 (Maxwell et al., 2021), and an overall accuracy between 85% to 99% (mean 94%). The GSP product is publicly available on the GeoService of the Earth Observation Center[2].

*Data availability: Global SnowPack datasets are available through the EOC GeoService:* https://geoservice.dlr.de/web/maps/eoc:gsp:daily

*Author contributions.* Conceptualization: AD SR. Data curation: AD SR. Formal analysis: SR. Investigation: AD SR. Writing: AD SR

*Competing interests.* I declare that I have no competing interests

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
