# Peer review of "Brief Communication: Daily, gap-free snow cover information based on a combination of NPP VIIRS and MODIS data"

_EGUsphere, 2025_

## Referee Comment (RC2)

**General comments**

The authors present an update to the "Global SnowPack" binary snow presence dataset to incorporate VIIRS NPP snow cover products alongside MODIS (MOD10A1 and MYD10A1) products. In addition to reading the Brief Communication, I had the benefit of reading the first reviewer's comments and the authors' responses. I agree with many points from the first reviewer and appreciate that the authors plan to address those concerns.

While the validation results presented in Figures 2 and 3 are helpful, readers may also be interested in how the updates have changed the Global SnowPack dataset from the previous version. As one example, was the dataset between 2000-2012 (when only MODIS was available) also resampled to 371 m resolution? If yes, how do the two versions compare during this period? Even if extensive validation data are not available, a relative comparison could give helpful information to potential data users.

**Specific comments**

Line 32-33: It is unclear which dataset the Landsat calibration/validation is applied to. If it's for the existing MODIS/VIIRS products, please provide a reference. If it's for the updated Global SnowPack data, please state this more explicitly.

Line 35: delete "has been prepared"

Line 54: Please define the F1 metric and include a sentence describing why was this metric was selected to determine the NDSI threshold.

Figure 2: Please note the different y-axis limits in the figure caption.

Figure 3/Lines 82-85: I guess the general idea for this dataset is that incorporating VIIRS provides more data coverage while not significantly decreasing the accuracy and recall presented in Figure 2. In the text, it would be helpful to potential data users to describe the additional cloud-free pixels in terms of additional snow-covered area. For example, the inclusion of an extra 10% of land pixels in the summer may not make much of a difference if those pixels are over parts of the planet where snow is not expected during that time of year. Could you give some examples of additional snow covered area that this new dataset captures that the others do not?

Figure 3: The caption references colors that are not actually present in the figure.

Line 66: This is the first time "snow line elevation information" has been presented – please provide more information here.

---

## Author Response (AR1)

**Answers to reviewer 1:**

We appreciate very much the time and effort that the reviewer took to review our manuscript and provide us such valuable feedback. Before addressing the comments in detail, I want to just mention that because of the "Brief Communication" format, we were very limited in terms of space and number of figures/tables.

**To answer the two major comments:**

*1. the introduction does not explain the stated what's the added value of your GSP dataset with respect to other similar gap-free products (e.g. Modis gap filled M[O-Y]D10A1F) or IMS (https://doi.org/10.7265/N52R3PMC). A review of other snow cover extent products is clearly behind the scope of the paper, but there could be the room for briefly highlighting why one should prefer this dataset.*

Our answer: Compared to the NSIDC products MOD10A1F and MYD10A1F, several steps are taken to fill gaps. Instead of only looking at the preceding days, a 3-day interpolation (in both directions) and a topographic interpolation are also performed. Furthermore, the information from three sensors (Terra and Aqua MODIS and NPP VIIRS) is initially combined daily, which fills most of the gaps. The Interactive Multisensor Snow and Ice Mapping System (IMS) also offers a combination of different sensors, but not at this resolution (IMS has a 1 km resolution) and not globally (IMS only covers the Northern Hemisphere). We will make sure to add this information to the manuscript to make it clearer to the reader.

*2. The evaluation of the GSP was performed using in situ data and Landsat products. Regarding Landsat the only result is one sentence "The comparison with Landsat resulted in an F1 score of 0.94". The discussion of the results is very limited. In the conclusion the authors wrote that the "uncertainties can easily be estimated and included in an uncertainty layer", however there is no further indication on how to actually estimate this uncertainty and the product does not provide such layer. The lack of information regarding Landsat products and the minimalist description of the results could prevent potential users from using this dataset.*

Our Answer: Again, the reason we only included this one, very short statement about the Landsat validation is the limited space we had with only 4 pages. After discussing this issue with the editor, we will extend the manuscript now slightly to include more information about the Landsat comparison, the location of the test sites, and details about the accuracy assessment. The quality layer is created during the process of cloud interpolation. Depending on the method that is used to fill the gap, and the duration of the cloud cover in every single pixel location, the quality of that pixel can be calculated. We conducted extensive tests and simulations of artificial cloud cover into clear scenes, interpolating them, and determining the uncertainty connected to the different interpolation steps and cloud cover durations. These simulations help us to construct a quality layer which is provided together with our Global SnowPack dataset, and a short description about that will be included in the revised manuscript. Here is also a figure illustrating the location of the Landsat validation sites (quality is not the best due to restrictions uploading figures in the answers, but hopefully enough to get an idea):

[Figure]

**Minor comments and our answers:**

- *In the absence of a user manual, more details about the dataset would be useful for users: is it operational ? What's the latency ? In which grid is it distributed ?*

Answer: It is operational, provided daily with a 3-4 day time lag to Real time conditions. The grid is WGS 84 (EPSG: 4326), covering the entire globe. The spatial resolution is 12 arc-seconds (approximately 370 meters)

- *L28: An illustration (a sample) of the dataset would be a nice addition.*

Answer: We unfortunately cannot add any more figures, but the latest product is always available under the following link: https://geoservice.dlr.de/web/maps/eoc:gsp:daily

- *L32: « Landsat data » Which Landsat ?*

Answer: All available Landsat 5, 7, 8, covering all continents. There will be a more detailed description about that in the revised manuscript

- *L33: « A total of 381 Landsat scenes » : the geographical and temporal distribution of these tiles is briefly given later ("distributed worldwide and covering all ecozones and seasons"). It would be more straightforward to give this information in paragraph 2 (and not 2.1). Also, additional information about these tiles could be useful: are mountain regions and high latitudes covered? In which years were these images acquired? .*

Answer: The Landsat tiles cover all regions, also mountains worldwide. The whole Landsat part will be revised in the manuscript to make sure we incorporate all these comments.

- *L33-35: The sentence is confusing. A reader cannot understand how the reference data were prepared. In (Koeheler et al. 2022) the NDSI is defined between a blue and a NIR band, in SnowPex exercise there are few different methods used for snow cover detection. Can you clarify?*

Answer: The full descritpion of how the Landsat data was processed is available in Koehler et al 2022. Essentially, all methods described by SnowPex have been tested and we used the one that produces the best results for the Alps and Pyrenees Mountains. That could of course mean that in other regions, another approach would have worked slightly better, but we assume that our classification is accurate. We will

add more information about this to the then extended Landsat part of the revised manuscript

- *L36: GMTED2010 is a multi resolution dataset. Which spatial resolution was used?*

Answer: We use the 7.5 arc-second resolution and resampled it to 12 arc-seconds using bilinear resampling.

- *L39-41: it is unnecessary to repeat here the motivation of this work especially in a "Brief communication".*

Answer: Will be modified in the revised manuscript

- *L43: The daily combination step could potentially benefit from satellite viewing angle information. Plenty of evaluations show how MODIS/VIIRS information is more reliable closer to nadir [1], [2], [3].*

Answer: Thanks a lot for this advice. We are thinking to include this aspect in a new update. For now, if more than one observation is cloud-free, selection is pixel-based, initially on the quality flag in the "Basic_QA" layer. If the values are equally good, the prioritization is based on VNP, MYD, and then MOD – since the focus is on better spatial resolution, and MYD will be added later for better lighting (the equator crossing is at 1:30 p.m. local time).

- *L55: What is the 3000 in the parenthesis next to MODIS products ?*

Answer: The 3000 in the parentheses indicates that these are the most appropriate NDSI thresholds after the MOD10A1 or MYD10A1 products were upscaled from 2400 to 3000 pixel size (to match the spatial resolution of VIIRS). But since this is not relevant for the reader, we will remove it from the revised manuscript

- *L66-77: should be in the Data & Method section*

Answer: we will move it to the data and methods

- *L70 (Figure 2 legend) : MOD10 identifier is a bit confusing here. MOD10 refers to MODIS/Terra products (e.g. MOD10A2). Please refer to the products ID (MOD10A1,MYD10A1,VNP10A1).*

Answer: we will make sure to clarify this and include the full product ID

- *L70 (Figure 2): maybe a logarithmic scale for the snow depth threshold would be more appropriate.*

Answer: We will look into that and test whether a logarithmic visualization is better

- *L80: « This accuracy….Figure 2b). » Can you elaborate? Accuracy reaches a maximum during summer months and decreases in the winter, meaning that accuracy is lower when the snow cover is greater at global scales. Any explanation?*

Answer: The reason for the higher accuracy in summer is the reduced cloud cover and particularly the much smaller snow cover dynamics: Snow field and glaciers which exist in summer are much more stable, so interpolating a cloud gap – even if it persists many days – will not introduce many errors. In winter, when much more snow cover is present, more clouds are present, and the dynamics are much higher, cloud gaps and polar darkness can last many days and therefore introduce higher uncertainties. The polar night gap itself can be 90 days or more – usually not in very dynamic regions when it

comes to snow cover, but still, errors happen more frequently here than in summer. In addition, lighting conditions are worse in winter, which makes snow detection more challenging.

- *L81 Again inconsistent notation MOD10/VIIRS.*

Answer: Will be addressed

- *L84 (Figure 3) Can you specify for which year or years was this plot computed?*

Answer: This figure is a compilation of all years in which MODIS and VIIRS were available (since January 19, 2012). The orange and purple parts of the figure show the information gain from the pixels added by VIIRS.

**Answers to reviewer 2:**

*General comments The authors present an update to the "Global SnowPack" binary snow presence dataset to incorporate VIIRS NPP snow cover products alongside MODIS (MOD10A1 and MYD10A1) products. In addition to reading the Brief Communication, I had the benefit of reading the first reviewer's comments and the authors' responses. I agree with many points from the first reviewer and appreciate that the authors plan to address those concerns. While the validation results presented in Figures 2 and 3 are helpful, readers may also be interested in how the updates have changed the Global SnowPack dataset from the previous version. As one example, was the dataset between 2000-2012 (when only MODIS was available) also resampled to 371 m resolution? If yes, how do the two versions compare during this period? Even if extensive validation data are not available, a relative comparison could give helpful information to potential data users.*

**OUR ANSWER:**

We thank the reviewer for their time and the effort they have put into the review. We know how much work such reviews are and sincerely appreciate the support. The general comment about the years prior to VIIRS is an important one: Yes we do provide these years in 371 m resolution, because we want to make it as easy as possible for the user to work with consistent time series. It is true that the change of resolution in these early years does not produce any advantage or improvement. It increases file size, which is a downside, but it allows for smooth analysis of the product and this aspect was most important for us. We have not looked into the relative comparison as it will only affect the border between snow-covered and snow-free areas, having minor effects which will eventually even out when considering whole time series or larger areas of interest. We will try to evaluate this effect though and make sure to introduce some information about it in the revised version of the manuscript

**Specific comments**

Line 32-33: It is unclear which dataset the Landsat calibration/validation is applied to. If it's for the existing MODIS/VIIRS products, please provide a reference. If it's for the updated Global SnowPack data, please state this more explicitly.

**OUR ANSWER:** The NDSI threshold for classifying binary snow cover from the NSIDC MOD10 and VIIRS datasets needed to be determined. Because the NSIDC products come with a layer containing NDSI, a decision for the binary classification needs to be made. Although there are plentiful suggestions in literature which threshold to use, we performed our own comparison with 381 Landsat scenes. The location of the 381 Landsat tiles cannot be illustrated due to the lack of space, but we made sure to distribute these reference Landsat scenes evenly over the planet. They were classified as snow/no snow according to Koehler et al and SnowPEX suggestions, and were then compared with the NDSI layers from the MOD10 and VIIRS datasets. This led to individual NDSI thresholds for both sensors.

Line 35: delete "has been prepared"

**OUR ANSWER::** Will be deleted

Line 54: Please define the F1 metric and include a sentence describing why was this metric was selected to determine the NDSI threshold.

**OUR ANSWER:** F1 was used as it is a widely used metric in remote-sensing based classifications, and because it is particularly useful when validating imbalanced classes. We will include a reference to the publication "Maxwell, A. E., Warner. T.A., Guillén L., (2024): Accuracy Assessment in Convolutional Neural Network-Based Deep Learning Remote Sensing Studies—Part 2: Recommendations and Best Practices. Remote Sens. 2021, 13(13), 2591; https://doi.org/10.3390/rs13132591"

Figure 2: Please note the different y-axis limits in the figure caption.

**OUR ANSWER**: Will be fixed

Figure 3/Lines 82-85: I guess the general idea for this dataset is that incorporating VIIRS provides more data coverage while not significantly decreasing the accuracy and recall presented in Figure 2. In the text, it would be helpful to potential data users to describe the additional cloud-free pixels in terms of additional snow-covered area. For example, the inclusion of an extra 10% of land pixels in the summer may not make much of a difference if those pixels are over parts of the planet where snow is not expected during that time of year. Could you give some examples of additional snow covered area that this new dataset captures that the others do not?

**OUR ANSWER:** We only process the Global SnowPack in areas where we expect snow. Therefore, an additional 10% is actually already very helpful. In the end, the exact location of the additional cloud-free information varies drastically. We could try to

produce a map, highlighting the areas where such additional data is available most often. Because of the limitation of the amount of figures we will have to see what is possible though.

Figure 3: The caption references colors that are not actually present in the figure.

**OUR ANSWER:** Thanks a lot for pointing that out. We changed the colors several times and this must have slipped our attention.

Line 66: This is the first time "snow line elevation information" has been presented – please provide more information here.

**OUR ANSWER:** True, and unfortunate: This is one of the steps we use to interpolate the Global SnowPack and is described in more detail in the GSP paper which is referenced. It is correct though that the mentioning of the Snow Line elevation method here is confusing for anybody not familiar with the GSP product. We will modify this part.